# Study of the Variation of Phenolic Acid and Flavonoid Content from Fresh *Artemisiae argyi Folium* to Moxa Wool

**DOI:** 10.3390/molecules24244603

**Published:** 2019-12-16

**Authors:** Min Li, Xin Chai, Luyao Wang, Jing Yang, Yuefei Wang

**Affiliations:** Tianjin Key Laboratory of TCM Chemistry and Analysis, Tianjin University of Traditional Chinese Medicine, Tianjin 301617, China; lm7911826@163.com (M.L.); chaix0622@tjutcm.edu.cn (X.C.); W1224967197@163.com (L.W.)

**Keywords:** *Artemisiae argyi Folium*, moxa stick, moxa wool, phenolic acids, flavonoids

## Abstract

*Artemisiae argyi Folium* (AAF) is a popular herbal medicine that is always employed in moxa sticks and by oral dosage in clinical use. Less attention has been paid to nonvolatile compounds as active compounds, such as phenolic acids and flavonoids. In this study, we focused on the variation rule of phenolic acids and flavonoids in the various transformations of *Artemisiae argyi Folium*. Using the established ultra-performance liquid chromatography (UPLC) method with an excellent methodology under “spider-web” mode, six phenolic acids and three flavonoids were simultaneously quantified in fresh and drying *Artemisiae argyi Folium* as well as in moxa wool and residue. Some interesting phenomena about the variation rule of phenolic acids and flavonoids were uncovered. First, a sharp increase was observed in the detected compounds’ content as the moisture gradually decreased, when fresh *Artemisiae argyi Folium* was exposed to sunlight and ambient or high temperature. Nevertheless, the increased phenolic acids were subjected to high temperature, leading to obvious degradation under oven-drying (60 °C and 80 °C). Second, a wide content distribution was revealed for the detected compounds in *Artemisiae argyi Folium* from different habitats, especially rutin, caffeic acid, chlorogenic acid, jaceosidin, eupatilin, and cryptochlorogenic acid. Third, accompanied by the elevated ratio of *Artemisiae argyi Folium*/moxa wool, the detected compounds conspicuously decreased in moxa wool and the correspondingly removed powder as residue. Importantly, a greater variation was found in moxa wool. Our findings contribute to the optimization of the drying process, the quality evaluation of the various transformations of *Artemisiae argyi Folium*, and the distinctive characterization of moxa wool produced at different ratios of *Artemisiae argyi Folium*/moxa wool.

## 1. Introduction

As a popular herbal medicine, *Artemisiae argyi Folium* (AAF) originates from a composite family [1], has the functions of removing dampness and relieving pain, removing blood stasis and swelling, dredging collaterals and warming meridians, and promoting blood circulation, and is conducive to treating diseases caused by wind chill [2,3,4]. As an important raw material, AAF is always used for manufacturing moxa sticks and for oral dosage in clinical application, such as aerosol, tablet, and oral liquid [5]. As an important medicinal form of AAF, the moxa stick is prepared from moxa wool with different ratios of AAF/moxa wool in the market, such as 1:1, 5:1, 10:1, and 15:1. AAF is made up of epidermis, veins, mesophyll, and petiole. It is deduced that the vein is the main composition of moxa wool, whereas the epidermis, mesophyll, and petiole are pulverized into powder as residue from the crushing process. By taking 5:1 (AAF/moxa wool) as an example, 5 kg AAF are processed repeatedly to remove powder as residue (about 4 kg) and about 1 kg moxa wool is obtained by the grinder. Along with the elevated ratio of AAF/moxa wool, the price increases correspondingly. However, the variation of phenolic acid and flavonoid content in moxa wool produced at different ratios of AAF/moxa wool has been barely reported.

As for the chemical compounds in AAF, except for volatile compounds as effective compounds, triterpenes, flavonoids, tannins, and phenolic acids are identified as the main chemical composition [6,7]. Accumulating evidences have suggested that phenolic acids and flavonoids are potentially pharmacological components. After retrieving information resources, we observed that phenolic acids have been reported to possess antimicrobial [8], antioxidation [9], and anti-inflammatory activities [10]. On the other hand, flavonoids exert antimutagenicity [11], inhibition of gastric ulcer [12], and anti-inflammatory and anticancer activities [13].

As important effective substances, phenolic acids and flavonoids have not been the focus of research. Only a few articles have reported a quantitative analysis of detected compounds by HPLC-UV carried out for at least 55 min [14,15]. Few publications have reported the simultaneous determination of phenolic acids and flavonoids, which we studied in less time. For multicomponent analysis, it is very important to efficiently extract compounds with different polarity. The “spider-web” mode presented by our group is qualified to appraise multi-index comprehensive evaluations, which have been successively applied in the optimization of extraction conditions, the screening of quality markers (Q-markers), and the quality evaluation of Chinese material medicine [16,17,18]. Compared to the reported methods for multi-index comprehensive evaluations, the “spider-web” mode possesses advantages about visualization and extensible capacity for factors of interest [17].

In addition, some important questions have not been investigated, such as transformation behavior of the detected compounds induced by the drying process, general survey of AAF from different habitats, and discrepancy of the detected compounds in moxa wool produced at different ratios of AAF/moxa wool in previous studies. Solutions to these issues will contribute to clarifying the relationship between quality and drying process and the habitats, as well as characteristics about compound distribution in moxa wool produced at different ratios of AAF/moxa wool, resulting in the quality assurance of AAF and moxa sticks.

Focusing on these issues, nine compounds were quantified in our study in 32 min by ultra-performance liquid chromatography coupled with a diode array detector (UPLC-DAD) according to the different appearances, namely fresh AAF, drying AAF, moxa wool, and residue, including cryptochlorogenic acid (cCGA), chlorogenic acid (CGA), caffeic acid (CA), isochlorogenic acid A (iCAA), isochlorogenic acid B (iCAB), isochlorogenic acid C (iCAC), rutin (RUT), jaceosidin (JAC), and eupatilin (EUP). The optimization of the extracting method was successively carried out in a simple and visual manner using the “spider-web” mode for the detected compounds with inverse polarity. The established method allowed us to discover the drying method’s significant influence on the surprising production of hydrophilic compounds (cCGA, CGA, RUT, iCAB, iCAA, and iCAC). Additionally, the considerable fluctuation of the detected compounds was investigated in AAF from different habitats. Moreover, a systematic investigation was performed on the appearance and the detected compounds of moxa wool produced at different ratios of AAF/moxa wool, hoping to elucidate the distinctive characteristics.

## 2. Results

### 2.1. Optimization of Extraction Condition and Analytical Method Validation for Quantitation of Nine Compounds

When optimizing the extraction conditions, the greatest challenge we encountered was how to balance the hydrophilic compounds, cCGA, CGA, CA, RUT, iCAB, iCAA, and iCAC, and the lipophilic compounds, JAC and EUP, as flavonoid aglycones. The extracting solvents had an inverse effect on compounds with different properties in AAF powder, leading to asynchronization of the extracting efficiency. To optimize a simple and efficient extracting method, the “spider-web” mode was preferably employed for optimizing the extraction conditions. The extracting efficiency of the detected compounds was primarily taken into account, including cCGA, CGA, CA, RUT, iCAB, iCAA, iCAC, JAC, and EUP. To express this concisely, the content of the target compounds was assigned as C′_m−k_ and C_m−k_, which were obtained, respectively, using an ultrasonic extracting method with the assistance of heating (60 °C) or not, and correspondingly divided by their highest content to give E′_m−k_ and E_m−k_. Calculation formulas were expressed as follows: *k* stands for the different compounds, namely cCGA, CGA, CA, RUT, iCAB, iCAA, iCAC, JAC, and EUP, and *m* is denoted as the 50% (I), 60% (II), 65% (III), 70% (IV), 75% (V), and 100% (VI) methanol aqueous solution:(1)$$E_{m-k}^{\prime }=C_{m-k}^{\prime }/C_{k}^{\prime }\left(max\right)$$
(2)$$E_{m-k}=C_{m-k}/C_{k}^{\prime }\left(max\right)$$

Using the extracting method by 50% methanol aqueous solution as an example, E′*_I__–cCGA_*, E′*_I__–CGA_*, E′*_I–CA_*, E´*_I–RUT_*, E′*_I–iCAB_*, E′*_I–iCAA_*, E′*_I–iCAC_*, E′*_I–JAC_*_,_ and E′*_I–EUP_* were employed to establish nine dimensions of the “spider-web” mode (*p′*_i_) for ultrasonic extracting by heating (60 °C), and E*_I–cCGA_*, E*_I–CGA_*, E*_I–CA_*, E*_I–RUT_*, E*_I–iCAB_*, E*_I–iCAA_*, E*_I–iCAC_*, E*_I–JAC_*_,_ and E*_I–EUP_* were used to set up nine dimensions of the “spider-web” mode (*p*_i_) for ultrasonic extraction, all of which are displayed in Figure 1. The calculation formulas were employed to calculate the shaded areas of the “spider-web” mode [19]. The bigger the shaded area, the closer it was to the most optimal extraction condition. The angle between the two dimensions was tagged as α′ and α (α′ = α = 360°/n, n = 9), respectively.
(3)$$S^{\prime }=\frac{1}{2}\sin\alpha^{\prime }\left(\sum_{i=1}^{n-1}p_{i}^{\prime }\;\cdot \;p_{i+1}^{\prime }+p_{n}^{\prime }\;\cdot \;p_{1}^{\prime }\right)$$
(4)$$S=\frac{1}{2}\sin\alpha \left(\sum_{i=1}^{n-1}p_{i}\;\cdot \;p_{i+1}+p_{n}\;\cdot \;p_{1}\right)$$

Based on the systematical investigation, an important phenomenon was observed, that is, the ultrasonic extracting method can significantly improve the extracting efficiency of the detected compounds with the aid of heating (60 °C), allowing for the successful optimization of extracting conditions. We proved that a 65% methanol aqueous solution was the satisfactory solvent to extract the compounds of interest with a “spider-web” area at 2.54, after a thorough screening of different solvents (50%, 60%, 65%, 70%, 75%, and 100% methanol aqueous solution). Furthermore, optimization was performed to achieve the best extracting ability, such as soaking time, ratio of solvent/material, extracting temperature, extracting time, and so on. In general, the sample powder was ultrasonically extracted by a 65% methanol aqueous solution for 30 min at a ratio of 100:1 (solvent/material) by heating (60 °C) after soaking for 1 h.

The study focused on the quantitation of nine compounds in the various transformations of AAF (fresh and drying AAF, and moxa wool and residue), and the optimized extracting method paved the way for methodological validation, the detailed results of which are shown in Table 1. Excellent linear relationship was achieved for all nine detected components in the tested range. The overall LOD and LOQ values were less than 0.12 and 0.35 μg/mL, respectively. Satisfactory results were obtained for intra- and inter-day precisions, repeatability, and stability with all RSDs below 3.0%. In addition, the average recoveries were 91.58–100.9% for the investigated compounds with all RSDs below 2.98%. Consequently, the established method can be successfully applied to determine the target compounds from the various transformations of AAF.

### 2.2. The Surprising Production of Phenolic Acids and Flavonoids in AAF Derived from the Drying Process

As reported, the accumulation of bioactive compounds is primarily limited by enzymes, which undertake the biosynthesis of secondary metabolites from primary metabolites in medicinal plants during the growth process [20]. This is tightly regulated by medicinal plants themselves and by environmental factors [21,22]. Recently, an increasing number of interesting transformations involving bioactive compounds have been discovered during processing, such as *Salviae miltiorrhizae* and *Fructus Psoraleae* [23,24,25]. Few reports have been published about the production of phenolic acids and flavonoids in AAF derived from the drying process.

In this study, we came across the surprising phenomena that only trace amounts of phenolic acids and flavonoids were detected in fresh AAF, in which the hydrophilic compounds cCGA, CGA, RUT, iCAB, iCAA, and iCAC were quantified at 11.31, 45.76, 8.956, 14.55, 22.84, and 33.38 μg/g, respectively. However, in dry AAF, a large number of hydrophilic compounds were unbelievably produced. It was speculated that hydrophilic compounds were generated during the post-harvest drying process. In order to reveal how hydrophilic compounds were being produced, different drying methods were tested, including drying in the shade (27–29 °C), sunshine drying (38–48 °C), and oven-drying (60 and 80 °C).

As illustrated in Figure 2, in order to scientifically express the production process of hydrophilic compounds, the calculated content of detected compounds was adjusted to a final content of dry AAF by taking the close relationship between moisture content and the content of detected compounds into account. CA is not presented in Figure 2, since it was almost not detected in this study. When fresh AAF (Nanyang, Henan, China) was subjected to the different drying conditions, the content of detected compounds (except for JAC and EUP) soared along with the decreased moisture, which began to obviously increase at about 30% moisture content. From 27 to 80 °C, the drying temperature had a significant effect on the production of hydrophilic compounds. Exposed to drying in the shade (27–29 °C), fresh AAF gradually lost moisture in 56 h. We observed that iCAB, iCAA and iCAC began to be dramatically produced at about 22 h, whose final content grew by almost 55- to 128-fold. A similar phenomenon was also recorded in sunshine drying (38–48 °C). Notably, the time was advanced at 1 h, at which point the obvious transformation of hydrophilic compounds occurred. Moreover, along with the increase of drying temperature, it took less time to detect the increase in content of the target compounds. Compared with drying in the shade and sunshine drying, a different trend was outlined about the transformation of hydrophilic compounds in oven-drying (60 and 80 °C). The content of hydrophilic compounds culminated at 1.5 h and 1 h, respectively, when fresh AAF was dried at 60 and 80 °C. After extending the drying time, the target compounds degraded to reach a balanced level, whose final content rose by about 3- to 25-fold. We speculated that instability resulted in the degradation of the hydrophilic compounds (phenolic acids and rutin).

As for triggering factors that produced the target compounds, valuable information was obtained by searching the library for information on this topic. It suggested that phenolic acids were mainly transformed from polysaccharides by biosynthesis process initiated by enzymes, or phenolic compounds linked to polysaccharides via ester and ether binding were released by acidic hydrolysis under the stress from the drying process [24,25,26]. In order to substantiate the transformed mechanism of the target compounds, a 65% methanol aqueous solution with different concentrations of hydrochloric acid (0, 0.2, 0.4, and 0.6 moL/L) was added to fresh AAF (Tangxian, Hebei, China) and used to ultrasonically extract the compounds of interest for 30 min at 60 °C. The detected results in our study proved that the different concentrations of hydrochloric acid had little influence on the extraction of phenolic acids, suggesting that the bound phenolic compounds to polysaccharides were not present in fresh AAF. Therefore, the potential enzyme in fresh AAF was suddenly triggered at a critical point, such as the decreased moisture content, which can generate the production of the compounds of interest.

In general, taking energy consumption and accumulation of active compounds into account, we recommend drying in the shade or sunshine drying as the preferred method. At present, sunshine drying is always popular in primary processing.

### 2.3. Quantitative Analysis of Phenolic Acids and Flavonoids in AAF from the Different Habitats

This newly verified UPLC-DAD method was successfully applied in the simultaneous quantification of six phenolic acids and three flavonoids in AAF. The representative chromatograms of the mixed reference compounds and the sample solution of AAF are shown in Figure 3A,B.

From the analyzed results, the components’ content greatly fluctuated in 16 batches of AAF from different habitats, namely Hubei, Hebei, Henan, and Liaoning. The nine quantified components reached 0.2690–1.029 mg/g for cCGA, 1.339–7.055 mg/g for CGA, 0.04744–0.3890 mg/g for CA, 0.03224–0.6610 mg/g for RUT, 1.229–2.696 mg/g for iCAB, 3.456–10.84 mg/g for iCAA, 2.156–5.922 mg/g for iCAC, 0.09804–0.7709 mg/g for JAC, and 0.1679–1.498 mg/g for EUP. A detailed description of the boxplot is shown in Figure 4. The content distribution of hydrophilic and lipophilic compounds was especially dispersed with RSDs between 25.77 and 70.93%. Many factors can bring about this phenomenon, such as the place and time of collection, the drying process, etc. As for the place of collection, the geo-authenticated habitat of AAF is Qichun (Hubei, China). The compounds’ quantified results barely proved that the content of the detected compounds in AAF from Qichun exceeded that from other habitats. Currently, the main producing areas of AAF are located in Henan Province (China) [27], where AAF is manufactured with higher contents of cCGA, CA, iCAB, iCAC, JAC, and EUP. From our study, the drying process is considered to be the most important influencing factor, resulting in the obvious variation of the detected compounds, especially for cCGA, CGA, iCAB, iCAA, and iCAC.

### 2.4. Discrepancy of Phenolic Acids and Flavonoids in Moxa Wool Produced at Different Ratios of AAF/Moxa Wool

Moxa wool is rolled up into a moxa stick, which is manufactured from AAF by a grinding process. Moxa wool of 1:1, 5:1, 10:1 and 15:1 (AAF/moxa wool) ratios is usually available in the market. Despite the big price differentiation, there is no feasible approach to distinguishing the moxa wool produced at different ratios of AAF/moxa wool. To overcome this challenge, we attempted to improve this situation by studying the different moxa wool according to appearance and ingredient distribution. The typical chromatograms of moxa wool and the correspondingly removed powder as residue are displayed in Figure 3C,D.

In our study, along with the increase in the AAF/moxa wool ratio (1:1, 5:1, 10:1 and 15:1), the appearance of moxa wool varied from dark brown to a grayish yellow, indicating that changes had occurred in the material composition. Detailed information is displayed in Figure 5. According to the quantitative results of the compounds of interest in moxa wool and the corresponding powder as residue, an obvious downward trend was observed with the increased ratio of AAF/moxa wool for the hydrophilic compounds (cCGA, CGA, CA, RUT, iCAB, iCAA, and iCAC), the content of which dropped by 58.67–84.06%. Moreover, the content of the hydrophilic compounds in residue was higher than that in the corresponding moxa wool. For the lipophilic compounds, JAC and EUP, a slight decline was observed, decreasing by about 25% from 1:1 to 15:1 (AAF/moxa wool). Because of the asynchronous decline of hydrophilic and lipophilic compounds, the content ratio of the hydrophilic compound (C_i_, i represents the different compounds) and the lipophilic compound (C_J_ and C_E_, J and E, respectively, stand for JAC and EUP) was further used to distinguish the different moxa wool. Along with the increasing of AAF/moxa wool, C_i_/C_J_ and C_i_/C_E_ descended stepwise and dropped by 44.26–78.54% and 45.83–79.15%, respectively. The results are shown in Figure 6. From this study, we concluded that the most notable characteristic is that the higher the ratio of AAF/moxa wool, the more obvious is the depletion of hydrophilic compounds in moxa wool, which sheds light on how to distinguish moxa wool produced at different ratios of AAF/moxa wool.

## 3. Materials and Methods

### 3.1. Reagents and Materials

Acetonitrile was purchased from Sigma-Aldrich (St. Louis, MO, USA), which was adequate for HPLC analysis. Formic acid was purchased from Shanghai Aladdin Bio-Chem Technology Co. Ltd. (Shanghai, China). Water used in this study was purified by a Milli-Q water purification system (Millipore, Billerica, MA, USA). CGA and RUT were purchased from the National Institute for Food and Drug Control (Beijing, China). cCGA, CA, iCAB, iCAA, iCAC, JAC, and EUP were obtained from Shanghai Yuanye Bio-Technology Co. Ltd. (Shanghai, China).

Sixteen batches AAF were collected from the different habitats. Three batches of moxa wool of different ratios of AAF/moxa wool (1:1, 5:1, 10:1 and 15:1) and the correspondingly removed powder as residue were manufactured by Hebei Chunkai Pharmaceutical Co. Ltd. (Hebei, China). Sample information is summarized in Appendix A. The fresh AAF was collected from the Nanyang cultivation base (Henan, China) and the Tangxian cultivation base (Hebei, China). The task of sample collection was carried out by Hebei Chunkai Pharmaceutical Co. Ltd. (Hebei, China). All of the samples were deposited in the Institute of Traditional Chinese Medicine, Tianjin University of Traditional Chinese Medicine (Tianjin, China).

The samples were prepared from fresh AAF by the authors of this study, and were used to study the transformation of compounds using the different drying methods (drying in the shade, sunshine drying, and oven-drying). The samples were collected at 0, 5, 10, 22, 44, and 56 h under drying in the shade (27–29 °C). The samples were gathered at 0, 1, 2, 3, 4, 5, and 6 h in sunshine drying (temperature of land surface between 38 and 48 °C). Under oven-drying (60 °C), the samples were taken at 0, 0.5, 1.5, 2.5, 3.5, and 5.5 h. Similarly, the samples were obtained at 0, 0.5, 1, 1.5, 2, and 2.5 h in oven-drying (80 °C). In order to homogenize fresh AAF, fresh leaves were sliced into small pieces about 3 cm wide before drying. Three parallel samples were conducted at each sampling point. Each fresh sample was about 40 g. The weight of every collected sample was measured to calculate the moisture content.

### 3.2. Preparation of Standard Solution

Nine reference compounds, namely cCGA, CGA, CA, RUT, iCAB, iCAA, iCAC, JAC, and EUP, were accurately weighed, dissolved, and diluted with methanol to obtain individual stock solutions at a final concentration of 1 mg/mL, which were employed to prepare a mixed reference solution at a concentration of 40.12 μg/mL for cCGA, 99.10 μg/mL for CGA, 10.09 μg/mL for CA, 20.06 μg/mL for RUT, 99.20 μg/mL for iCAB, 204.4 μg/mL for iCAA, 198.6 μg/mL for iCAC, 39.48 μg/mL for JAC, and 59.28 μg/mL for EUP. Then, the mixed reference solution was serially diluted by 50% methanol aqueous solution to give seven different concentrations for building the calibration curve.

### 3.3. Preparation of Sample Solution

Accurately weighed sample powder (0.5 g) was immersed by adding 50 mL 65% methanol aqueous solution about 1 h in a conical flask (100 mL). Subsequently, it was ultrasonically extracted for 30 min at 60 °C and cooled to ambient temperature. The sample solution was centrifuged at 14,000 rpm for 10 min and then injected into UPLC-DAD for analysis.

### 3.4. UPLC-DAD Analysis

Using a Waters ACQUITY UPLC system, UPLC analysis was undertaken on an ACQUITY UPLC BEH Shield RP18 column (2.1 × 100 mm, 1.7 μm) at 50 °C in gradient elution with mobile phase consisting of 0.2% formic acid aqueous solution (A) and acetonitrile (B). The gradient elution implemented at 0.3 mL/min was as follows: 5–18% B in 0–6 min, 18–22% B in 6–10 min, 22–31% B in 10–12.5 min, 31–43% B in 12.5–18.5 min, 43–60% B in 18.5–23 min, 60–74% B in 23–24 min, and 74–90% B in 24–32 min. The detection wavelength was set at 360 nm. Injection volume was 2 μL.

### 3.5. Methodological Validation

The analytic method established in this study was implemented to validate linearity, LOD, LOQ, precision (intra- and inter-day), stability, and recovery testing. The standard curves were drawn based on the peak area (*y*) of the analytes and the corresponding seven different concentrations (*x*) in duplicate. Determination of LOD and LOQ was finished with a standard solution at S/N about 3 and 10, respectively. In order to test the intra- and inter-day precision, injections of the same sample solution six times on the same day and on three consecutive days were conducted, respectively. Six samples were processed and analyzed to confirm the repeatability. The stability was validated by replicate injection of sample solution at 0, 2, 4, 6, 8, 10, and 12 h, respectively.

### 3.6. Data Analysis

The “spider-web” mode was performed by Excel 2016.Lnk software. The line graphs and boxplot were plotted using GraphPad Prism 5.01 software (GraphPad software Inc. San Diego, CA, USA).

## 4. Conclusions

Confirmed by the well-established UPLC-DAD method, quantitative analysis of nine compounds derived from phenolic acids and flavonoids was performed for different forms of AAF, including fresh and drying AAF, and moxa wool and residue. Sunshine drying was adopted as the ideal method for drying AAF based on the evident accumulation of interesting compounds. Moreover, drying AAF presented highly fluctuating quantitative results about the detected compounds, which was probably caused by the different habitats and drying methods. Moxa wool was processed from the drying AAF, from which the hydrophilic compounds showed a more conspicuous decline with the increased ratio of AAF/moxa wool. The results may pave the way for a guarantee of AAF quality and the identification of moxa wool produced at different ratios of AAF/moxa wool.

## Figures and Tables

**Figure 1 molecules-24-04603-f001:**
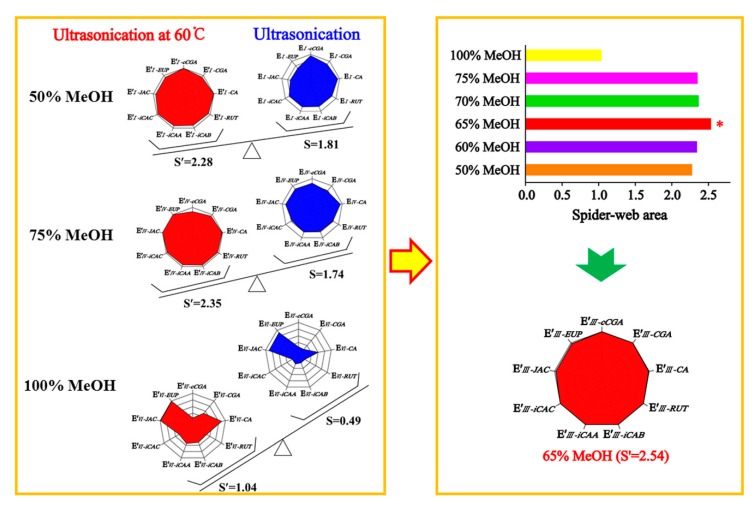
Optimization of the extracting method for *Artemisiae argyi Folium* (AAF) using the “spider-web” mode.

**Figure 2 molecules-24-04603-f002:**
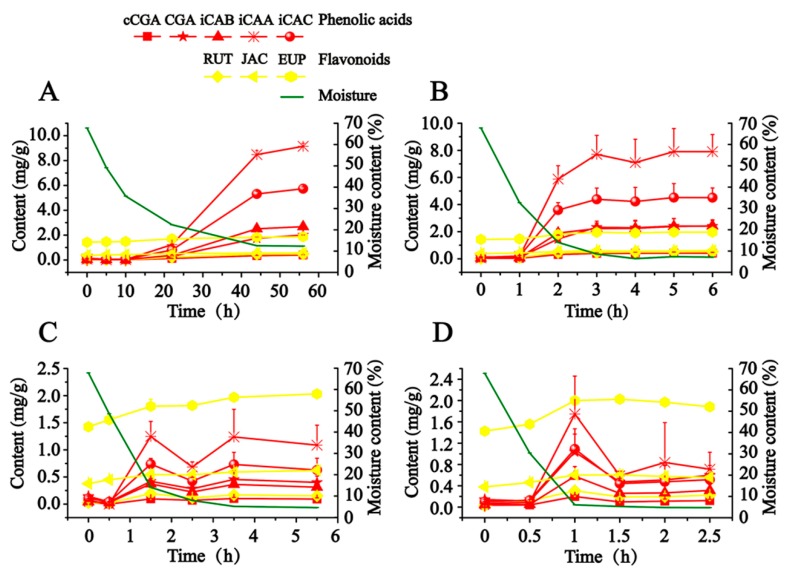
The dynamic variations of phenolic acids, flavonoids, and moisture in AAF exposed to drying in the shade (**A**), sunlight drying (**B**), oven-drying at 60 °C (**C**), oven-drying at 80 °C (**D**).

**Figure 3 molecules-24-04603-f003:**
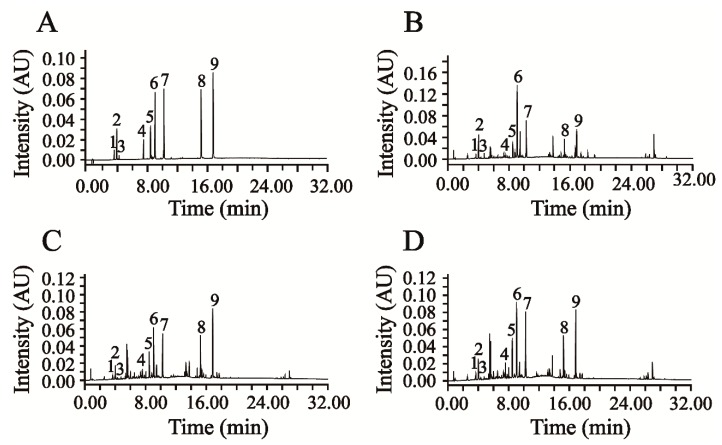
The representative chromatograms of mixed reference compounds (**A**), and sample solution of AAF (**B**), moxa wool (**C**) and residue (**D**) (1. cCGA, 2. CGA, 3. CA, 4. RUT, 5. iCAB, 6. iCAA, 7. iCAC, 8. JAC, 9. EUP).

**Figure 4 molecules-24-04603-f004:**
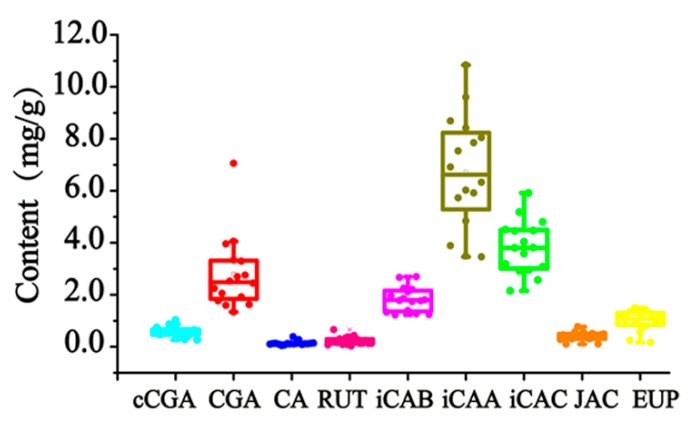
The boxplot of the detected compounds in AAF from the different habitats.

**Figure 5 molecules-24-04603-f005:**
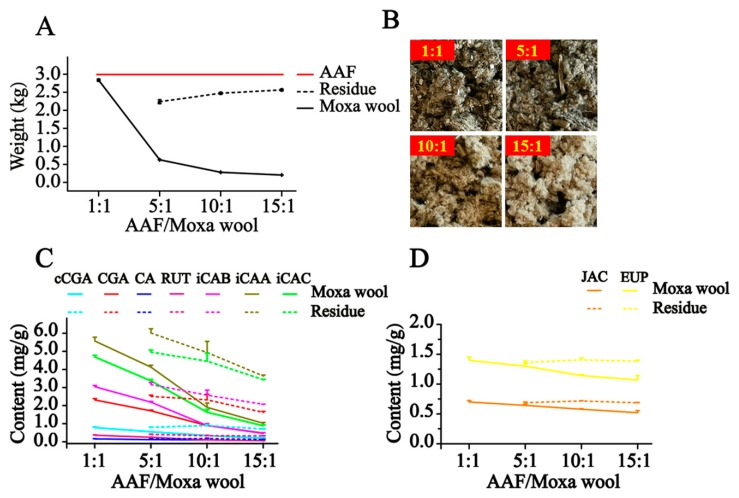
The dynamic curves of weight of moxa wool and residue produced at different ratios of AAF/moxa wool (**A**), and the photograph of moxa wool (**B**), the dynamic curves of content variation of hydrophilic compounds (**C**) and lipophilic compounds (**D**) in moxa wool and the correspondingly removed powder as residue along with the increased ratio of AAF/moxa wool.

**Figure 6 molecules-24-04603-f006:**
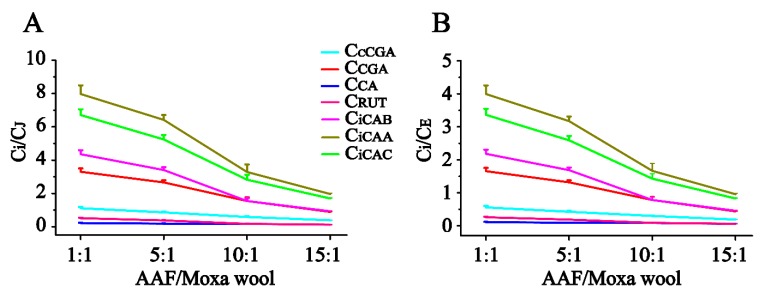
The dynamic curves of content ratio of hydrophilic compounds to jaceosidin (**A**) and hydrophilic compounds to eupatilin (**B**) in moxa wool along with different ratios of AAF/moxa wool.

**Table 1 molecules-24-04603-t001:** Summary results of linear regression, LODs, LOQs, precision, repeatability, stability, and recovery for the tested compounds in AAF.

Compounds	Linear Regression			LODs (μg/mL)	LOQs (μg/mL)	Precision RSD (%)		Repeatability (*n* = 6, RSD, %)	Stability (*n* = 6, RSD, %)	Recovery (*n* = 6, Mean ± SD, %)
	Regression Equation	*r* ^2^	Linear Range (μg/mL)			Intra-Day	Inter-Day			
cCGA	*y* = 2855.4 *x* − 67.44	0.9999	0.63–40.12	0.0697	0.2090	0.44	1.79	1.28	0.64	99.89 ± 0.72
CGA	*y* = 3477.9 *x* − 993.4	0.9999	1.55–99.10	0.0573	0.1720	0.14	1.56	1.57	0.22	91.58 ± 2.25
CA	*y* = 4552 *x* − 59.12	0.9999	0.16–10.09	0.0526	0.1577	0.47	2.83	1.06	0.60	94.54 ± 2.44
RUT	*y* = 9688.3 *x* + 67.94	0.9999	0.31–20.06	0.0348	0.1045	0.56	2.29	1.80	1.04	94.49 ± 2.35
iCAB	*y* = 4229.5 *x* − 500.07	0.9999	1.55–99.20	0.0574	0.1722	0.25	2.56	1.85	0.77	98.40 ± 2.02
iCAA	*y* = 4635.7 *x* − 3574.6	0.9999	3.19–204.40	0.1183	0.3549	0.16	2.70	1.99	0.68	100.94 ± 3.00
iCAC	*y* = 5453.4 *x* − 5633.3	0.9999	3.10–198.60	0.1149	0.3448	0.26	3.00	1.94	0.40	95.42 ± 1.98
JAC	*y* = 24,812 *x* −2665.7	0.9999	0.62–39.48	0.0228	0.0685	0.20	0.62	1.88	0.17	98.28 ± 1.54
EUP	*y* = 23,205 *x* − 3124	0.9999	0.93–59.28	0.0343	0.1029	0.15	1.60	1.14	0.13	96.58 ± 2.33

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
