# Peer review of "Study of the Variation of Phenolic Acid and Flavonoid Content from Fresh Artemisiae argyi Folium to Moxa Wool"

_molecules, 2019, doi:10.3390/molecules24244603_

Round 1
Reviewer 1 Report
My questions, corrections, suggestion, etc., are in the enclosed text of manuscript.

Author Response
Comments from reviewer 1, Please view the attachment.

Reviewer 2 Report
In several paragraphs is used the abreviation et al for other phases more than those related with refererences, can you explain why?
Author Response
Comments from reviewer 2, Please view the attachment.

Reviewer 3 Report
The manuscript entitled “Study the characteristics about variation of phenolic acids and flavonoids from fresh Folium Artemisiae Argyi to moxa wool” is devoted to the implementation of novel ‘spider-web’ approach for extraction parameter optimization and HPLC-UV profiling of Q-markers in leafs and herbal medicines that contain Artemisiae argyi. Although having potent therapeutic potential, Artemisiae species are not comprehensively studied especially in the aspect of variation of the hydrophilic constituents. Therefore, the part of the article that describes phenolic acids and flavonoids dynamic variation in the drying processing of Artemisiae argyi leaves is highly valuable for the future researches in this field. However, I cannot recommend this manuscript to be published in present form. My major concern is the organization of this paper and insufficiency of some of its parts. First, the Introduction is 1/3 discussion of the previous work (lines 34-46) and 2/3 the describtion of the present manuscript (lines 47-66). I believe that the Introduction can be enhanced by the important information related with moxa wool chemistry and some data about alternative approaches to the ‘spider-web’ system developed by the authors. From this, the next two important questions step into focus: 1) an eligibility of using ‘spider-web’ approach to compare different extraction systems should be discussed with respect to the existing approaches for multi-component extraction optimization; 2) moxa wool as a part of the investigated FAA/moxa wool TCM should be introduced to a reader. In section 2.4, it is clearly shown that the contents of the nine compounds of interest from Artemisiae argyi Folium are decreasing, while FAA/moxa wool ratio increases from 1 to 15. If I understood correctly, moxa wool is prepared from related species Artemisia vulgaris which may also contain similar Q-markers, and the different decline rates are connected to the different initial concentrations of these constituents in the ingredients. It will be helpful to see free of FAA moxa wool Q-marker contents in the Figure 5C and 5D. In any case, it is unclear how the processing of moxa wool with Artemisiae argyi Folium could affect the chemical composition of the resulting product? In what way the “residue” is produced? All of these questions are arising because of the Materials and Methods section being too shallow: no technological steps of moxa wool production were described. It should also be pointed out, when the authors calculate the content by adjusting it to dry weight. Similarly, the “spider-web” approach could be described in a more detailed way, i.e. how one should determine the angle for two studied cases α and α’? Is it always could be defined as α = 360/n (number of components/dimensions?) The authors claim (lines 70-72) that solvent had inverse effect on compounds of interest, so what are the solvents that were used? In the studied case, 65-75% methanol affords almost close to 1.0 extraction efficiency (Figure 1) for all nine compounds, thus no asynchronization case could be observed from this data.
Author Response
Comments from reviewer 3, Please view the attachment.

Round 2
Reviewer 1 Report
Think it is Ok now.
Author Response
Thank you very much for your comments.
Reviewer 3 Report
The manuscript has been improved by the authors.
Remaining problems and questions that appeared from the changes made are:
1) Please indicate in one of the sections that you calculate the contents adjusted to dry weight of the samples.
2) Please insert the decribtion about the above mentioned leaf veins, mesophyll and petiole crashing processing for better understanding.
3) It is unclear what exactly the authors mean by the word "characteristics" in the Lines 1-2: Study the characteristics about variation of phenolic acids and flavonoids content; and Lines 60-61: Possesses characteristics about visualization and extensible capacity for interesting factors.
4) Line 63: "Also, some important questions were not be illuminated", should be were not illuminated.. in previous studies.
5) It is unclear what does the therm "methodologic validation" exactly mean?
